# Translating neural signals to text using a Brain-Computer Interface

**Janaki Sheth**
Department of Physics and Astronomy
University of California, Los Angeles
Los Angeles, CA 90095
`janaki.sheth@physics.ucla.edu`

**Ariel Tankus**
Department of Neurology and Neurosurgery
Tel Aviv University
Tel Aviv, Israel
`arielta@post.tau.ac.il`

**Michelle Tran**
Department of Neurosurgery
University of California, Los Angeles
Los Angeles, CA 90095
`metran@mednet.ucla.edu`

**Nader Pouratian**
Department of Neurosurgery
University of California, Los Angeles
Los Angeles, CA 90095
`npouratian@mednet.ucla.edu`

**Itzhak Fried**
Department of Neurosurgery
University of California, Los Angeles
Los Angeles, CA 90095
`ifried@mednet.ucla.edu`

**William Speier**
Department of Radiology
University of California, Los Angeles
Los Angeles, CA 90095
`speier@ucla.edu`

## Abstract

Brain-Computer Interfaces (BCI) may help patients with faltering communication abilities due to neurodegenerative diseases produce text or speech by direct neural processing. However, their practical realization has proven difficult due to limitations in speed, accuracy, and generalizability of existing interfaces. To this end, we aim to create a BCI that decodes text directly from neural signals. We implement a framework that initially isolates frequency bands in the input signal encapsulating differential information regarding production of various phonemic classes. These bands form a feature set that feeds into an LSTM which discerns at each time point probability distributions across all phonemes uttered by a subject. Finally, a particle filtering algorithm temporally smooths these probabilities incorporating prior knowledge of the English language to output text corresponding to the decoded word. Further, in producing an output, we abstain from constraining the reconstructed word to be from a given bag-of-words, unlike previous studies. The empirical success of our proposed approach, offers promise for the employment of such an interface by patients in unfettered, naturalistic environments.

## 1 Introduction

Neurodegenerative diseases such as amyotrophic lateral sclerosis (ALS) restrict an individual's potential to fully engage with their surroundings by hindering communication abilities. Brain-Computer Interfaces (BCI) have long been envisioned to assist such patients as they bypass affected pathways and directly translate neural recordings into text or speech output. However, practical implementation of this technology has been hindered by limitations in speed and accuracy of existing systems [4]. Many patients rely on devices that use motor imagery [10], or on interfaces that require them to individually identify and spell out text characters such as the "point and click" cursor method

33rd Conference on Neural Information Processing Systems (NeurIPS 2019), Vancouver, Canada.

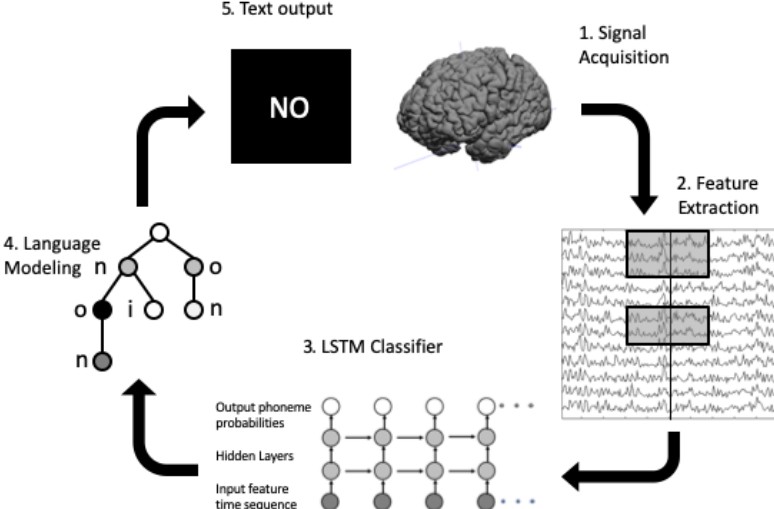

Figure 1: Overview of neural signals translation to text. 1) Signals are recorded using depth electrodes implanted based on clinical need. 2) Signal features are selected for each time point based on spectral analysis. 3) A bLSTM creates probability distributions over phonemes at each time point. 4) Probabilities are smoothed and domain knowledge is incorporated using a probabilistic automaton traversed using a particle filtering algorithm. 5) The highest probability word is chosen as the output.

[11, 12]. Despite significant work in system optimization, inherent limitations in their designs render them significantly slower than spoken communication.

To address these shortcomings, several studies are using electrocorticography (ECoG) and local field potential (LFP) signals [2]. These invasive approaches provide superior signal quality with high temporal and spatial accuracy. Previous work attempted translation to continuous phoneme sequences using invasive neural data [8]; however, despite their reported higher translation speed, their applications are limited to a reduced dictionary (10-100 words). Other design choices meant to enhance phoneme classification capitalize on prior knowledge of the target words, hindering their generalization to unmodified scenarios. Additionally, a recent study synthesized speech using recordings from speech cortex. Though the authors demonstrate partial transferrability of their decoder amongst patients, their accuracy is again limited to selection of the reconstructed word by a listener from a pool of 25 words and worsens as the pool size increases [3].

Thus, establishing the capability of these approaches to generalize to unconstrained vocabularies is not obvious and has to our knowledge not yet been studied. Here, we present the performance of a two-part decoder network comprising of an LSTM and a particle filtering algorithm on data gathered from six patients. We provide empirical evidence that our interface achieves an average accuracy of 32% calculated against a full corpus, i.e. one encompassing all feasible English words that can be formulated using the entire set of phonemes uttered by a patient, thus marking an important, non-incremental step in the direction of viability of such an interface.

## 2 Methods

The overall system consists of five steps as detailed in Fig.1.

### 2.1 Experimental Design

During the study, subjects were asked to repeat individual words ("yes", "no"), or monopthongal vowels with or without preceding consonants. During each trial, they were instructed which word or string to repeat and were then be prompted by a beep followed by a 2.25 second window during which they spoke. A trial thus consists of each such repetition. The number of these trials varied between subjects based on their comfort, ranging from 55 to 208. Number of phonemes per subject

consequently fluctuated between 8 (3 consonants, 5 vowels) to 16 (11 consonants, 5 vowels). The sampling rate of these recordings was 30 kHz. Before further processing, electrodes determined to have low signal-to-noise ratio (SNR) were removed. Criterion for electrode removal was either that its time-series signal was uniformly zero, or that it contained artifacts that were atleast one order of magnitude larger than the mean absolute signal amplitude. Correspondingly, $\sim$ 8-9 % of channels were eliminated from further analysis. We elucidate in our previous work [9] the localization and relevance of the remaining electrodes as pertains to the different parts of speech they encode.

## 2.2 Feature Selection

In order to include as input to our network classifier differential information stored in the neural signals about production of various phonemes, an experiment was designed that mapped power in spectral bands of these recordings to the underlying phoneme pronunciation.

Each recording was divided into time windows from -166.67 to 100 ms relative to onset of the speech stimuli. Labels [0,1] were assigned respectively to the corresponding audio signal: [silence, consonant/vowel]. The power per band is pre-processed by z-scoring and down sampling it to 100 Hz. This then acts as an input to a linear classifier which we train using early-stopping and coordinate descent methods. To additionally ensure that the classifier can identify the silence after completion of a phoneme string, we performed training over 100 ms post speech onset, but test the features captured by the classifier over 333.33 ms, since most trials end within this time period.

While previous studies have used bands upto high gamma (70-150 Hz) for all speech uniformly, our results show that for several of our subjects, vowels are delineated by high spectral bands (> 600 Hz) and consonants by low ones (< 400 Hz). For individual subject's values we refer the reader to [9].

## 2.3 LSTM Model Description

The first part of our decoder is a stacked two-layer bLSTM. We use a bLSTM due to its ability to retain temporally distant dependencies when decoding a sequence [7]. Furthermore, our analysis reveals that while a single-layer network can differentiate between phonemic classes such as nasals, semivowels, fricatives etc.; a two-layer model can distinguish between individual phonemes. We train this model with an ADAM optimizer to minimize weighted cross-entropy error, wherein the weights are inversely proportional to phoneme frequencies. Finally, we evaluate the decoder using leave-one-trial-out; for each time point in the test trial the recurrent network outputs probability distributions across all phonemes in the subject's dataset. Implementation was using Pytorch [1].

## 2.4 Language Model

A language model is used to apply prior knowledge about the expected output given the target domain of natural language. In general, such a model creates prior probability distributions for the output based on the sequences seen in a corpus that is reflective of the target output. In this study, word frequencies were determined using the Brown corpus [5] translated into phonemic sequences using the CMU Pronouncing Dictionary [14]. The phoneme prior probabilities were determined by finding the relative frequency of each phoneme in the resulting corpus. To find probabilities of sequences of phonemes, these priors may be conjoined using the nth-order Markov assumption to create an n-gram model. While such models are able to capture local phonemic patterns, they allow for sequences that are not valid words on the language. A probabilistic automaton (PA) creates a stronger prior by creating states for every subsequence that starts a word in the corpus [13]. Each state then links to every state that represents a superstring that is one character longer (Figure 2). This graphical structure accounts for the possibility of homophones by keeping a list of such words associated with each node along with their relative frequency in the text corpus. Here we would like to reiterate that our model solely derives from the Brown corpus, and hence as such is patient agnostic.

## 2.5 Temporal smoothing

Laplacian smoothing is applied to the LSTM model output so that phonemes that were not seen during training are assigned a non-zero probability. To the resulting distributions we then apply the language model using a subject-dependent temporal algorithm. Specifically in this study, we employ a particle filtering (PF) method [13]. PF estimates the probability distribution of sequential outputs by

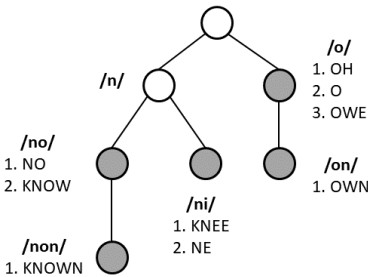

Figure 2: Example probabilistic automaton based on phonemes: /n/, /o/, and /i/. Shaded nodes represent possible terminations for a word. Lists of words for a given node correspond to homophones.

creating a set of realities (called particles) and projecting them through the model based on observed data and the language model [6]. Each particle contains a reference to a state in the model, a history of previous states, and an amount of time that it is going to remain in the current state. Distribution of states occupied by these particles represents an estimation of the true probability distribution.

When the system begins, a set of P particles is generated and each is associated with the root node of the language model. At each time point, samples are drawn from the proposal distribution defined by the transition probabilities from the previous state.

$$x_t^{(L)} \sim p(x_t \mid x_{t-1}^{(L)}) \tag{1}$$

The time that the particle will stay in that state is drawn from a distribution representing how long the subject is expected to spend speaking a specific phoneme. At each time point, the probability weight is computed for each of the particles using,

$$w_t^{(L)} \propto w_{t-1}^{(L)} p(y_t \mid x_t^{(L)}) \tag{2}$$

The weights are then normalized and the probability of possible output strings is found by summing the weights of all particles that correspond to that string. The system keeps a running account of the highest probability output at each time. The effective number of particles is then computed.

$$P_{eff} = \frac{1}{(\Sigma_i (w_t^{(L)})^2)} \tag{3}$$

If the effective number falls below a threshold, $P_{thresh}$, a new set of particles are drawn from the particle distribution. This threshold was chosen empirically to be 10% of the total number of particles. Sensitivity analysis varying this value did not have a significantly affect results. Further at each time point, the amount of time for a given particle to remain in a state is decremented. Once that reaches zero, the particle transitions to a new state in the language model based on probability $p(x_t \mid x_{0:t-1})$.

## 3    Results

In our evaluation, output words were only considered correct if the phoneme sequence matched the labels and each phoneme overlapped at least partially with its respective audio label. Word accuracies varied between subjects, ranging from $54.55\%$ (subject 1) to $13.46\%$ (subject 2) (Table 1). On average, $32.16\%$ of trials were classified completely correctly and an additional $23.06\%$ had at least one phoneme match. Of the incorrect classifications $32.28\%$ produced incorrect words either because none of the output phonemes were correct or because the sequences did not align temporally with the audio signal. In the remaining $12.49\%$ of trials, the system did not detect speech signals, and produced an empty string as output.

## 4    Discussion

Each of the subjects in this study were able to communicate with significantly higher accuracy than chance. Nevertheless, the average word error rate seen in this study (67.8% on average) was higher

Table 1: Word level performance of each subject.

| Subject | $ACC_W$ (%) | Partial (%) | Incorrect (%) | Omission (%) | Baseline (Random chance) |
|---------|-------------|-------------|---------------|--------------|--------------------------|
| 1 | 54.55 | 12.73 | 20.00 | 12.73 | 2.36 |
| 2 | 13.46 | 21.15 | 46.15 | 19.23 | 0.52 |
| 3 | 23.08 | 35.75 | 36.65 | 4.52 | 0.28 |
| 4 | 29.73 | 18.92 | 32.43 | 18.92 | 1.27 |
| 5 | 31.43 | 24.57 | 33.71 | 10.29 | 0.76 |
| 6 | 40.72 | 25.26 | 24.74 | 9.28 | 0.74 |
| Average | 32.16 | 23.06 | 32.28 | 12.49 | 0.99 |

than the 53% reported in [3]. There were several important differences in these studies, however. The primary difference is that their system produced an audio output that required a human listener to transcribe into a word selection. Despite advances in machine learning and natural language processing, humans have superior ability to use contextual information to find meaning in a signal. Furthermore, that study limited classifications to an output domain set of 50 words, which is generally not sufficient for a realistic communication system.

While this study makes a significant addition to existing BCI literature in terms of its avoidance of the traditional bag-of-words approach, our accuracies are lower than those reported in ERP-based BCI studies [12]. Moreover, in order for a BCI system based on translating neural signals to become a practical solution, improvements need to be made either in signal acquisition, machine learning translation, or user strategy. One approach could be to sacrifice some of the speed advantages by having users repeat words multiple times. While this would reduce communication speeds below natural speaking rates, it would still greatly exceed ERP-based methods, while increasing the signals available for classification which could improve system accuracy.

However, both this study and previous literature have primarily been concerned with decoding speech/text for patients with intact motor abilities. It is presently unclear how this would translate to intended speech. While the electrodes used in this study are inept to answer this question, given their majority location in the speech cortical areas [9], we suggest a plausible new experiment: teaching those who can't speak to rethink speech in terms of vocal tract movements. Using electrodes in the sensorimotor cortex [3] and continuous visual feedback of ground truth vocal tract movements for each phoneme's pronunciation, a subject's attention could be entrained to only the (intended or executed) motion of their vocal tract for covert and overt speech respectively. One can then test the transferability of state space models - latent variables comprising of different articulators and observed states corresponding to the time-varying neural signals - between the covert and overt behaviours to better understand and harness the physiological variability between the two to eventually translate current studies into potentially viable devices.

## 4.1 Limitations and Future Work

The language model used in this study was designed to be general enough for application in a realistic BCI system. This generality may have been detrimental to its performance in the current study, however, as language models based on natural language will bias towards words that are common in everyday speech. The current study design produced many words that are infrequent in the training corpus. As a result, the language model biased away from such outputs, making it almost impossible to correctly classify. Lastly, while the results presented in this study are promising, but they represent offline performance which does not include several factors such as user feedback.

## 5 Conclusion

The proposed system serves as a step in the direction of a generalized BCI system that can directly translate neural signals into written text in naturalistic scenarios. However, communication accuracies are currently insufficient for a practical BCI device, so future work must focus on improving these and developing an interface to present feedback to users.

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
