# OpenReview forum: "Translating neural signals to text using a Brain-Computer Interface"
_NeurIPS.cc/2019/Workshop/Neuro_AI — Real Neurons & Hidden Units @ NeurIPS 2019 Poster_

### Official Review · AnonReviewer1 · 2019-09-23
**Important BCI decoding topic; needs more explanation of study basics**

**Clarity:** 4

**Comment:**

This is a nice paper as is, and will be better if the authors can address my issues with the technical clarity. With some more effort given to formatting (like removing double-spacing of bibliography), there should be space for these improvements.

I would like to hear what the authors think about decoding speech, where the subject actually says a word, versus decoding intended speech, where the subject only imagines saying a word. Intended speech sounds more difficult to me, since there won't be any motor signals. This is an important challenge to overcome for someone with motor impairment. Is there any way to interpret the signals that this framework learns? Can you tell which electrodes are most relevant for the decoding task, and are these in speech or motor areas? This is probably something that could go into the discussion.

**Category:**

AI->Neuro

**Clarity Comment:**

The paper is generally well-written with correct grammar and good explanations. My main issues are with the technical clarity (see technical comments).

Small edits:
* L. 1, I'd add "may" before help
* L. 2, could strike "output" from "speech output"
* L. 33, "the speech cortex" -> "speech cortex"
* Ll. 39-42, this last sentence is a run-on. Also, it is missing a comma before "i.e."
* L. 48, use backticks `` for open quotes in Latex
* L. 60, missing a comma "Across multiple subject, vowels..."
* L. 63, add "The" to start of 1st sentence
* L. 64, "Futher" -> "Furthermore"
* Figure 1, typo "nodes"

**Evaluation:**

3: Good

**Importance:**

4: Very important

**Importance Comment:**

Decoding speech (or intended speech) from neural signals is an important problem with wide potential clinical applications. This study moves towards a more naturalistic setting by decoding a larger set of words than what was previously attemped. This is an important step towards a useful device.

**Intersection:**

5: Outstanding

**Intersection Comment:**

This is an excellent example of an AI method applied to an important neuroscience problem.

**Rigor Comment:**

The main issue I have with the paper lies in the description of the experiment. It is not clear what constitutes a "trial". What are the lengths of the time bins after computing the spectral power? This is presumably coarser than 30 kHz. What data are used for training versus testing, and are these random bins or non-interleaved segments of the total timeseries? From the description of the loss as cross-entropy, it sounds like the output target for the LSTM is phoneme identity. What is the total number of phonemes? It is unclear whether the language model built on top of this is actually trained on the subject data or the phoneme to word map is just a result of the Brown corpus + CMU dictionary. This needs to be made clear.

Similarly, it sounds like the goal of the study is to decode words from a very large corpus. However, the experimental design section states that the subjects only performed trials where they said "yes", "no", or phoneme strings which presumably don't map to real words. So what is the part where you try and decode their actual speech? Is it from observations of the rest of their hospital stay while they are speaking with family, friends, doctors, and nurses? Is this what's tested after only training on the phoneme string data? This should all be made explicit.

I found the level of detail in the LSTM section more than necessary, and the explicit parameters of the ADAM method and learning rate could be left out. (As is, the level of detail in this paper is not enough to make the study reproducible, but I am glad to see the code will be made available. With the code, you don't need these parameters in the text.) Instead, I would use the space gained to explain more the basic experimental setup and algorithm design.

On the other hand, there could be more detail about the smoothing + particle filter steps. Is the automaton model a Markov chain? Figure 1 doesn't add much in my opinion. You could add a lot more information by creating a diagram of how phoneme + automaton gives the word output. Adding some math like the PF update equations could make everything more precise.

**Technical Rigor:**

2: Marginally convincing

---

### Official Review · AnonReviewer3 · 2019-09-26
**BCI to read text from neural signal**

**Clarity:** 2

**Category:**

AI->Neuro

**Clarity Comment:**

The text is very brief in terms of experimental detail and methods. It is challenging to figure out what exactly the algorithm did and the advantage of this particular algorithm based on the results written.

**Evaluation:**

2: Poor

**Importance:**

4: Very important

**Importance Comment:**

This is a very important question. If the accuracy is high, an algorithm can increase the life quality of disabled or stroke patients.

**Intersection:**

3: Medium

**Intersection Comment:**

The authors tried to use RNN to encode LFP signal for speech recognition.

**Rigor Comment:**

The authors used a set a standard RNN (LSTM) network to use neural data (LFP) for word detection. It is a bit unclear how they select their features. It is also unclear how their algorithm perform (37% accuracy) compared to other algorithms. It is thus hard to judge the significance of result.

**Technical Rigor:**

1: Not convincing

---

### Official Review · AnonReviewer2 · 2019-09-27
**Important problem, interesting results for now**

**Clarity:** 4

**Comment:**

This is an interesting and promising approach.  The problem that is addressed is important and the methods seem sound. However, the authors limit their analysis to very simple stimuli (yes and no and some non-word sounds) which reduces the impact of their work (they motivate their approach as not being constrained to decoding out of a small pool.

**Category:**

AI->Neuro

**Clarity Comment:**

The paper is well written.

**Evaluation:**

4: Very good

**Importance:**

4: Very important

**Importance Comment:**

The goal behind the paper is very useful: being able to decode complex speech and not just choose an option from a limited pool. A solution which allows patients to communicate easily and rapidly would greatly improve their quality of life. Using a language model or any kind of recurrent model to use past brain activity to predict more accurately is a crucial direction and the authors are correct in pursuing it.

**Intersection:**

4: High

**Intersection Comment:**

This work is an example of using AI tools to decode brain activity and isn't about using AI as a model of what the brain is doing.  It is a nice illustration of the role that AI could achieve in that domain.

**Rigor Comment:**

The experiments appear convincing. The limited space doesn't allow a very deep understanding of all the procedures that were used.

**Technical Rigor:**

4: Very convincing

---

### Decision · Program_Chairs · 2019-10-02

Accept (Poster)